# Telerehabilitation in Children and Adolescents with Cystic Fibrosis: A Scoping Review

**DOI:** 10.3390/healthcare12100971

**Published:** 2024-05-08

**Authors:** Ezequiel Pessoa, Mara Ferreira, Cristina Lavareda Baixinho

**Affiliations:** 1Nursing School of Lisbon, 1600-190 Lisbon, Portugal; ezequiel.pessoa@esel.pt; 2Nursing Research, Innovation and Development Centre of Lisbon (CIDNUR), Nursing School of Lisbon, 1600-190 Lisbon, Portugal; maraferreira@campus.esel.pt; 3Centro Hospitalar Universitário Lisboa Norte, 1649-035 Lisbon, Portugal; 4Center for Innovative Care and Health Technology (ciTechcare), 2410-541 Leiria, Portugal

**Keywords:** cystic fibrosis, children, adolescents, telerehabilitation, scoping review

## Abstract

Cystic fibrosis requires regular monitoring and intervention by healthcare teams; despite that, adherence to therapeutic measures is less than desired. The evolution of technology has allowed much of the care provided in person to be replaced by a telehealth delivery model, but studies on telerehabilitation are scarce and dispersed. This scoping review aimed to identify which domains of rehabilitation intervention are mediated by information and communication technologies and how they are developed in the provision of care to children and adolescents with cystic fibrosis. The data collection was conducted in February and June 2023, following the three steps recommended by the JBI for this type of review: (1) the search was conducted in MEDLINE, CINAHL, Scopus, JBI, and Web of Science; (2) the bibliographic references obtained from the included articles were analysed; and (3) the grey literature was checked. The eligibility criteria were children and adolescents and rehabilitation interventions mediated by information and communication technologies. The five studies included in this review were subjected to analysis, and a narrative synthesis of the results was carried out. The interventions identified included physical exercise programs (60%), management of the therapeutic regimen (40%), and symptom control (40%). The information and communication technologies were web-based platforms, video games, and telephones. The use of telerehabilitation included face-to-face meetings to ensure participants performed the exercises correctly, monitor their response to exercise, and teach them how to avoid risky situations during home workouts. In all studies, exercise sessions were supervised by the participants’ parents or caregivers.

## 1. Introduction

Cystic fibrosis (CF) is a chronic, progressive, genetic, and hereditary disease and autosomal recessive disorder that impairs normal chloride secretions, resulting in thick secretions that primarily affect breathing [1,2]. Respiratory impairment results from the presence of thick mucus, which causes airway obstruction, infection, lung damage, and pulmonary failure, shortening life expectancy [3,4].

Given the chronic, progressive, and disabling nature of CF, the therapeutic regimen is demanding and is based on multiple treatments, most of which are carried out on a daily basis [5]. Two of the pillars are airway clearance techniques (ACTs) and exercise training, within the scope of rehabilitation.

Airway clearance techniques have a positive impact on lung function decline. They clear secretions, but also improve lung function, increasing the percent predicted forced expiratory volume in one second (ppFEV1) [3,6,7]. Regular participation in physical activity and involvement in structured exercise training programs have also been shown to improve exercise tolerance, lung function, energy levels, and quality of life and are strongly encouraged [8,9].

Because the disease is diagnosed close to birth or in early childhood [5], support from families is essential to optimize the physical and psychosocial development of children living with CF [10]. Furthermore, childhood and adolescence (age >6 and <18 years) are fundamental stages for the development of skills related to understanding the disease and its management. As school-age children expand their cognitive and linguistic skills and can better communicate their beliefs and expectations about illness, psychoeducational interventions addressing misinformation and unfounded fears or concerns are more effective [5]. At this stage, it is also essential to encourage children’s sense of ownership and control, in order to develop skills and self-efficacy related to self-management of the disease and collaboration throughout their lifespan [5]. All these efforts should extend into adolescence, a phase that is characterized by persistent movement towards increasing autonomy, and in which self-concept continues to be refined through experiences such as “trying on” different roles and behaviours that could have implications for health [5].

Despite the tremendous physical and psychosocial development that occurs in these phases of the life cycle, CF typically worsens during adolescence with the presence of more frequent symptoms, like cough and fatigue, and pulmonary exacerbations, increasing the illness burden [11]. Research has shown that over time, overall adherence to CF treatment worsens with age, when children become adolescents [12,13].

There are several reasons that may explain problems related to adherence to the therapeutic regimen. In addition to the complexity and demands of treatment already mentioned, other problems have been reported, such as the accessibility of care, particularly that of reference centres, where disease management allows for better results, due to their distance from many patients or simply due to a lack of availability [14,15]. Contact isolation measures between patients with CF to prevent the onset of chronic airway infections when seeking healthcare cannot be excluded from this equation either [16,17].

The coronavirus disease 2019 (COVID-19) pandemic has exacerbated some of these problems and posed unique challenges for clinicians, researchers, and other healthcare professionals involved with CF patients [1]. Many of the activities based on a traditional in-person service model were quickly replaced by a telehealth delivery model, in a rapid adaptation to respond to the health needs of this population [18].

Telehealth, defined as the use of information and communication technologies (ICTs) to support distance-based health interventions like the assessment, education, monitoring, and delivery of healthcare interventions [19,20], has ensured timely remote healthcare and managed many chronic illnesses, with documented effectiveness [21]. Technology-based care delivery models are here to stay as a way to solve several problems, old and current, in accessing the best healthcare [22]. It also has the advantage of allowing easier access to reference centres in places where accessibility is limited, by distance or by service availability [14,15]. But telehealth also brings several challenges to multidisciplinary intervention, which also includes intervention in respiratory rehabilitation [14,15] with the need to design and test the effectiveness of telerehabilitation interventions for these patients and their families.

### Objectives

To date, we have not found any systematic reviews, including scoping reviews, on the use of telehealth in the provision of rehabilitation care—telerehabilitation—in children and adolescents with CF. Therefore, the aim of this scoping review (ScR) is to identify which domains of rehabilitation intervention are mediated by ICT and how they are developed in the provision of care for children and adolescents with CF.

## 2. Materials and Methods

The study protocol was registered in the Open Science Framework (OSF) (DOI 10.17605/OSF.IO/B8TDW) with extended data at https://osf.io/wn2zd (accessed on 16 February 2023).

The protocol was designed according to the steps defined by the Joanna Briggs Institute (JBI) for the synthesis of evidence in a scoping review (ScR) [23]. It was structured as recommended by the Preferred Reporting Items for Systematic Reviews and Meta-Analyses extension for Scoping Reviews (PRISMA-ScR) checklist [24].

### 2.1. Study Design

After conducting an initial search in the various databases, we did not identify any systematic reviews, including scoping reviews, to answer the following research question: What interventions in the area of rehabilitation, mediated by ICT, are in use and how are they developed in cystic fibrosis management in children and adolescents?

Given the state of the art, the best type of systematic review to answer the research question is the ScR. A research strategy was implemented following the recommendations of the JBI 2021 and aimed to be as comprehensive as possible to answer the research question, based on the PCC mnemonic framework (Participants, Concept, Context) [23].

### 2.2. Eligibility Criteria

Each element of the acronym guided the definition of each specific inclusion criterion, presented in Table 1.

The following were considered for inclusion: primary studies, quantitative and qualitative studies, as well as secondary studies, specifically all types of literature reviews, guidelines indexed in databases, and any studies in the grey literature that could answer the research question. The studies were limited to full texts, in Portuguese, English, and Spanish, with a time limit of five years (2017–2022), due to the timeliness of the results and because exploratory research to identify studies showed that the research on the subject is recent.

### 2.3. Information Sources and Search Strategy

An initial exploratory study was carried out in February 2023 to identify the state of the art, check descriptors, and structure the study protocol. In June, a search was carried out, following the 3 steps recommended by the JBI for this type of systematic review [23]. First, an initial search was conducted in the MEDLINE (via PubMed) and CINAHL (via EBSCO) databases, using keywords built from natural language relative to the theme. This search allowed the identification of the words in the titles and abstracts, as well as the indexing terms used. A second search was carried out using the keywords and indexing terms identified in the previously included databases. The Boolean operators OR and AND were used to operationalize the search, and language filters were applied for full text, language, and time restriction. In this phase, the search was conducted in MEDLINE (via PubMed), CINAHL (via EBSCO), Scopus, JBI, and Web of Science. The descriptors were adjusted to each database, e.g., for Medline, MeSH terms were used; for CINAHL, subject headings; and so on.

After this stage, a search was conducted in the grey literature on websites about cystic fibrosis, in master’s and doctoral thesis repositories, and the bibliographic references obtained from the included articles were analysed. Table 2 shows the complete search strategy for the MEDLINE database.

The articles identified in each database/source were exported to Rayyan^®^. Two researchers independently screened the titles and abstracts (E.P., M.F.) according to the predefined inclusion criteria. A full-text evaluation of the retrieved studies was then carried out. Disagreements between reviewers were resolved by a third reviewer (C.L.B).

After this phase, the articles were read in full by each researcher, and this analysis was verified by the research team, increasing reliability.

### 2.4. Study Selection and Data Processing and Analysis

The researchers created an Excel table that was shared in the cloud to record the characteristics of the content extracted from the articles in the final bibliographic sample: identification of the title of the article/work; author(s), year of publication, and type of article; and objective(s), method, and main results/conclusions. Two reviewers (E.P. and C.L.B.) tested the extraction form in three sources to benchmark the decision-making process and ensure that relevant results were recorded, in line with Valaitis et al. [25].

The articles that answered the research question and met the inclusion criteria were subjected to analysis and a narrative synthesis of the results was carried out.

### 2.5. Ethical Issues

This study was carried out strictly following the study protocol to ensure its validity. The identification and referencing of the articles included in the bibliographic sample followed the recommendations of good academic and scientific practice. The extraction and analysis of data was carried out with evident respect for the research and results obtained by other researchers.

## 3. Results

The search strategy initially generated 425 results, which, after screening and assessment for eligibility, resulted in the inclusion of 5 studies for review. The PRISMA flow diagram [24] is presented in Figure 1. Specifically, after the initial elimination of two duplicate results, the two reviewers analysed the titles and abstracts of the remaining articles against the inclusion/exclusion criteria. Forty-two full-text articles were reviewed, once again, against the inclusion and exclusion criteria, and five were included for review. The references of all included articles were selected, although they did not generate more articles. Searches were also carried out in repositories of academic institutions, as well as in organizations that work in CF, and four articles were identified.

The studies were heterogeneous and showed some geographic dispersion. They were conducted in the United States [1], Spain [26], Türkiye [27], Greece [28], and Brazil [29].

The included studies had different designs (Table 3). Two were RCTs [26,27], one a quasi-experimental study [28], one a feasibility/prospective study [1], and the other a cross-sectional study [29]. In the RCTs, the researchers who evaluated patients before and after the intervention were blinded to the participants’ treatment allocations. All studies were conducted as home-based interventions.

Three of the included studies were conducted with children and adolescents [26,28,29], one with children [27], and one with adolescents [1]. The number of participants ranged from 10 [1] to 184 [29]. With the exception of one study that did not report data on gender [28], in three studies the participants were mostly female [1,27,29], ranging from 53.7% to 71.4%, and in one study the participants were mostly male with 52.5% [26]. Three studies were conducted with participants with a stable clinical condition [1,26,27], one did not consider clinical stability as an inclusion criterion [29], and in another there were no data on this issue [28]. In three studies, participants were recruited from reference centres for the treatment of CF [1,28,29]; in one, from CF associations [26]; and in one, from a paediatric pulmonology clinic [27].

In three of the included studies, telerehabilitation programs consisted of home exercise training programs [1,26,27], and two of them addressed issues of the daily treatment routine or the control of signs and symptoms associated with CF in the form of teleconsultations or telemonitoring [28,29].

The studies that focused on physical exercise training used different strategies to develop the interventions. In one, an active video game (AVG) was used as a training modality (EA SPORTSTM ACTIVE 2 from Nintendo Wii^TM^ platform), designed to include a combination of aerobic exercise, muscular strength exercise, body endurance, and flexibility [26]. The game was supervised by a virtual personal trainer and included a heart rate (HR) monitor to monitor daily exercise intensity and help patients to control their HR evolution. To increase patient adherence, weekly telephone check-ins were also provided. In another study, a trained researcher actively participated and supervised, via videoconferencing (web-based platform Zoom), an exercise training program based on a combination of high-intensity interval training, in the form of a letter game, and postural strengthening [27]. In another study, a personal trainer, via videoconference (Zoom), led, instructed, demonstrated, and encouraged participants to engage in a supervised resistance exercise training (RET) programme that consisted of whole-body exercise sessions, with the additional use of a set of adjustable-weight dumbbells, in individual sessions [1].

In these studies, it was possible to find information about the frequency, intensity, type of exercise, and duration of the programme. The frequency of the programs varied from three [1,27] to five [26] times a week. The exercise intensity was specified, according to the type of exercise performed, for example, 70–80% of the maximum HR for aerobic exercise [26] and 60% of one-repetition maximum for resistance exercise training [1]; one study did not specify these data [27]. Only one study specified the duration of the session, which varied between 30 and 60 min [26]. The programme duration ranged from 6 [26] to 12 weeks [1,27].

In two studies, games were used to implement the exercise programme: one used a video game [26] and the other a form of letter game [27]. In one, the exercises were also adapted to the age of the participants—≤12 years and >13 years [26].

These studies also described the security conditions underlying the development of the programme. In one study, the first sessions were held in person to ensure participants performed the exercises correctly, monitor their response to exercise, and teach them how to avoid risky situations during home workouts [27]. In two studies, rehabilitation sessions took place face-to-face, via videoconference, with the supervision of therapists [1,27]. In all studies, exercise sessions were supervised by the participants’ parents or caregivers.

The studies that focused on the implementation of routine teleconsultations and telemonitoring also used different strategies. One used a web-based platform (Skype^®^) or a telephone, depending on the patients’ availability, to address important issues of the daily treatment routine like treatment adherence, the proper use of prescribed drugs, the possibilities of performing techniques, and the use of physical therapy equipment for each patient individually [29]. The other based the intervention on telephone communication, dealing with questions about exacerbations, respiratory infections or symptoms, weight gain, medications, and treatment adherence [28].

## 4. Discussion

This scoping review allowed the mapping of telerehabilitation interventions, including telemonitoring, in children and adolescents with cystic fibrosis, identifying the intervention design, as well as the e-health modality, technological platforms, and other elements, used to delivere digital care.

Although the research strategy was comprehensive in answering the research question, the low number of studies identified stands out for showing that telerehabilitation is still in development and being adopted by reference centres for monitoring the disease, still without dissemination of the results of its implementation. Even so, the results corroborate the opinions of other authors, who suggest that supervised telerehabilitation with telemonitoring support is feasible and safe, without adverse events, and adherence is high, with a recruitment rate consistent with previous respiratory rehabilitation trials [30,31].

The information and communication technologies used for intervention were web-based platforms [1,27,29], video game consoles [26], and telephones [29]. Intervention monitoring was carried out by researchers via teleconsultation (telephone or web platform) [28,29] or by a heart rate (HR) monitor [26]. These virtual interventions eliminate the need for unnecessary in-person referrals to specialists, reduce wait times for specialist feedback, and eliminate the need for unnecessary travel [32]. They also simultaneously allow the timely identification of clinical changes, worsening of symptoms, and non-adherence to the therapeutic regimen [30,31,32].

The good adherence of children and adolescents to these programmes demonstrates their acceptability and allows us to predict the potential of e-health interventions for solving problems present in these chronically ill patients. Some studies have observed that adherence is unsatisfactory, especially in certain therapeutic modalities, such as nutritional guidelines, inhalation therapy, and respiratory physical therapy [1,7,9,33].

Researchers who implemented a video game intervention concluded that it can effectively improve exercise capacity, muscular strength, and quality of life in the short term in children and adolescents with CF [26]. Indeed, video games have been considered as a novel way of delivering healthcare interventions with the potential to motivate changes in health behaviours [34,35]. One systematic review that aimed to examine the effectiveness of game-based interventions on physiological outcome measures, as well as adherence and enjoyment, in subjects with chronic respiratory diseases observed that the use of videogame interventions, undertaken for several weeks, improved exercise capacity and other outcomes and was more enjoyable for subjects with chronic respiratory diseases [34].

Another study that used videogames with patients over 6 years old with CF reported that active video gaming in patients with CF, as well in healthy individuals, generated the required cardiorespiratory demand for moderate-intensity aerobic physical training. However, these effects depended on the game selection, with the games Wii Fit “Free Run” (Nintendo Wii) and “Just Dance 2015″ (Xbox One, song “Summer”) being the ones that produced greater responses [36]. More studies with children and adolescents are needed that include investigations of the effects in this age group.

The interventions implemented were home exercise training programs, and controlling the signs and symptoms and treatment adherence. Physical activity and exercise have numerous benefits in patients with CF, including improved FEV_1_, aerobic capacity, lung function, exercise capacity, and quality of life [37]. These patients show some difficulties in exercise, symptom management, and therapeutic adherence that are essential for controlling the disease, preventing exacerbations, and slowing progression [37,38]. Non-compliance with the therapeutic regimen has multiple clinical and economic consequences, with increased mortality or morbidity observed in non-compliant patients [38]. Further studies are needed to determine the effect of telerehabilitation in adherence to therapeutic regimes.

Although they mentioned that a reduction in hospital visits is an advantage of these interventions [32], the researchers did not evaluate the participants’ satisfaction in relation to this factor. We suggest that future studies explore this topic, especially because these patients are often exposed to invasive and painful procedures in the hospital, and the fact that they need to visit the hospital increases their fear and worry [39]. There is also a need to understand the meaning that children, adolescents, and their families give to the possibility of having the care at home and understanding their representations, beliefs, and behaviours [40], which can interfere with the delivery of and adherence to telerehabilitation. Although paediatric telerehabilitation has been considered feasible to provide clinical intervention, future research is needed to evaluate the impact of telerehabilitation services on patient care and its applications for the ongoing use of this delivery model [41].

Another recommendation is that studies include other non-pharmacological interventions in association with physical exercise, given the multicomponent nature of interventions in young people with CF.

This scoping review has limitations related to some methodological options, such as restrictions placed on the language and free access to full texts, which probably excluded some articles that answered the research question. It should also be noted that the studies are heterogeneous in terms of design, participants, and type of intervention.

## 5. Conclusions

This ScR was conducted in MEDLINE, CINAHL, Scopus, JBI, and Web of Science. Of 425 identified studies, 5 respected the eligibility criteria, 2 were RCTs, 1 was a quasi-experimental study, 1 a feasibility/prospective study, and the other a cross-sectional study.

Interventions within the scope of telerehabilitation included physical exercise programs (60%), management of the therapeutic regimen (40%), and control of symptoms (40%). In some studies, the use of telerehabilitation included face-to-face meetings to ensure participants performed the exercises correctly, monitor their response to exercise, and teach them how to avoid risky situations during home workouts. In all studies, exercise sessions were supervised by the participants’ parents or caregivers. The information and communication technologies used for intervention were web-based platforms, video game consoles, and telephones. Intervention monitoring was carried out via teleconsultation (telephone or web platform) or by a heart rate monitor.

## Figures and Tables

**Figure 1 healthcare-12-00971-f001:**
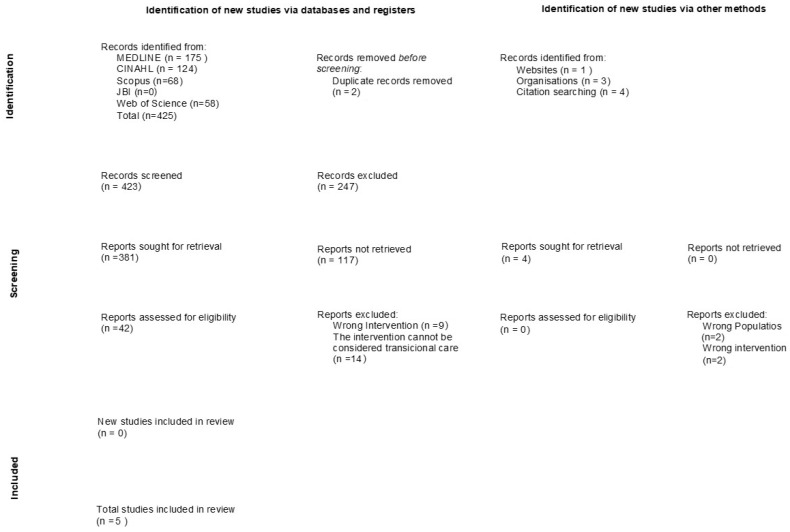
PRISMA 2020 flow diagram. Lisbon, 2023.

**Table 1 healthcare-12-00971-t001:** Eligibility criteria for bibliographic sample. Lisbon, 2023.

PCC	Inclusion Criteria	Exclusion Criteria
Population	The population of children (6–12 years old) and adolescents (13–18 years old) with CF	Adults and older persons
Concept	Rehabilitation interventions mediated by ICT (telerehabilitation). It was also predefined to accept studies about telemonitoring, teleconsultation, and e-nursing interventions if they were related to telerehabilitation interventions/programs.	‘Traditional’ face-to-face intervention
Context	Healthcare practice environments	Educational and/or social contexts

**Table 2 healthcare-12-00971-t002:** Search strategy for MEDLINE (via PubMed). Lisbon, 2022.

	Search Strategy	Total Articles
#1	(((((((((((((child[Title/Abstract]) OR (children[Title/Abstract])) OR (child*[Title/Abstract])) OR (“child”[MeSH Terms])) OR (adolescent[Title/Abstract])) OR (adolesc*[Title/Abstract])) OR (early adolescence[Title/Abstract])) OR (late adolescence[Title/Abstract])) OR (adolescence[MeSH Terms])) OR (adolescent[MeSH Terms]))) OR (adult children[MeSH Terms])) NOT (adult[MeSH Terms])) NOT (aged[MeSH Terms])	211,760 results
#2	((cystic fibrosis[Title/Abstract]) OR (CF[Title/Abstract])) OR (cystic fibrosis[MeSH Terms])	11,056 results
#3	(((((((((((digital health[Title/Abstract]) OR (e-health[Title/Abstract])) OR (e-nursing[Title/Abstract])) OR (Telerehabilitation[Title/Abstract])) OR (telemedicine[Title/Abstract])) OR (telenursing[Title/Abstract])) OR (Remote Consultation[Title/Abstract])) OR (Remote Sensing Technology[Title/Abstract])) OR (Mobile Applications[Title/Abstract])) OR (tech*[Title/Abstract])) OR (cellular telephone[MeSH Terms])) OR (telehealth[MeSH Terms])	372,525 results
#4	((((health[Title/Abstract])) OR (hospital[Title/Abstract])) OR (primary care[Title/Abstract])) OR (consultation[Title/Abstract])	704,492 results
#5	#1 AND #2 AND #3 AND #4	125

**Table 3 healthcare-12-00971-t003:** Telerehabilitation for children and adolescents with CF. Lisbon, 2023.

Study, Year, Country, Study Design, and Setting	Aim Sample	Telerehabilitation Intervention	Results/ Conclusions
[1] (2022) USA Feasibility/prospective study At home	Determine the feasibility of a home-based resistance exercise training (RET) programme in adolescents with CF and glucose intolerance using virtual personal training and the effects of the programme on glucose metabolism, pulmonary function, body composition, and physical fitness. Ten participants, age 10–18 ys. Mean age 15.80 ys (±2.20).	Technological requirements: web-based platform (Zoom); set of weight-adjustable dumbbells. Virtual personal training supervised via online videoconferencing. Intervention design: Home-based resistance training program supervised by a personal trainer who provided instruction, demonstration, and verbal encouragement in one-on-one sessions via live video calls. Presence of an adult parent/guardian during training sessions was required in case of injury or another emergency. Participants recorded the resistance training volume-loads (sets × repetitions × loads) for each exercise, which were totalled at the end of every session and every week. Frequency: 3 times/week on non-consecutive days. Intensity: Load was prescribed after completing a ten-repetition maximum (10RM) test for each exercise. Emphasis was placed on volume-based progression. The set number was increased every 3–4 weeks, from 1–4 sets, with 8–15 repetitions. Percentage of one-repetition maximum (%1RM) remained constant at ~60% 1RM. Duration: 12 weeks (36 sessions).	Telehealth-based RET is feasible in adolescents with CF and impaired glucose tolerance and elicits small yet favourable changes in insulin secretion, body composition, and exercise capacity.
[26] (2018) Spain RCT At home	Assess the effectiveness of a home exercise programme using an active video game (AVG) as a training modality for children and adolescents with CF. Total of 39 participants, age 7–18 years (ys). Experimental group n = 19, 10 males and 9 females, mean age 12.6 ys (±3.4); control group n = 20, 11 males and 9 females, mean age 11 ys (±3).	Technological requirements: video game console (Nintendo WiiTM platform) with an active video game (AVG)—EA SPORTSTM ACTIVE 2; heart rate (HR) monitor. Intervention design: The AVG activities were supervised by a virtual personal trainer and included a heart rate (HR) monitor to help patients control their HR evolution and monitor daily exercise intensity. The exercise activities were adjusted according to age (≤12 y and >13 y) to improve motivation among participants and the training load was increased every week. Initial training sessions were provided at specialized CF institutions to ensure that the participants performed the exercises correctly, monitor their exercise response, and teach them to avoid risky situations during training sessions at home. The subsequent training sessions were supervised by parents or caregivers at home. To increase patient adherence, a physiotherapist provided weekly telephone check-ins. After the training period, the AVG patients were instructed to continue their individualized exercise program using the same AVG at home for a 12-month follow-up period, with an exercise prescription of a minimum of 2 days per week, 20 min per session. Frequency: 5 days/week. Intensity: 70–80% maximal HR. Time: 30–60 min per session. Type: activities included running, squats, lunges, and bicep curls. Duration: 6 weeks.	A home-based program using AVGs can effectively improve exercise capacity, muscular strength, and quality of life in the short term in children and adolescents with CF. The effects of training on muscle performance and quality of life were sustained over 12 months.
[27] (2022) Türkiye RCT At home	Examine the effect of telerehabilitation on quality of life, depression, and anxiety levels in children with CF and their caregivers’ mood and anxiety levels. Total of 28 children 6–13 years old (and their caregivers). Telerehabilitation group n = 14, mean age 9.8 ys (±2.14); control group n = 14, mean age 10.0 ys (±1.64).	Technological requirements: web-based platform (Zoom). “The researchers that applied the exercise program were present during the whole session of the telerehabilitation and supervised and participated with the patient in person”. Intervention design: The exercise program was based on a combination of high-intensity interval training and postural strengthening and was prepared by a specialist physiotherapist. Frequency: 3 times/week. Duration: 12 weeks. Type: High-intensity interval training was performed using a letter game. A list containing 4-letter words with an exercise for each letter was given, and during the exercise program, each participant could perform the exercises by choosing a word in each session. E.g., first word, and the respective exercises: O: Take 20 steps sideways and hop back; K: Pretend to jump rope 20 times; U: Walk 15 steps like a horse; L: Do 3 somersaults. Another game was to write the name and each letter corresponded to an exercise, e.g., letter A, consisted of doing Jump up and down 10 times. Postural strengthening consisted of corner pectoral stretch, scapular retraction with external rotation, triceps brachii strengthening, biceps strengthening exercise, abdominal strength exercise, abdominal strength exercise, push ups, and back extensor strengthening.	A telerehabilitation approach that includes postural and aerobic exercises can help patients with CF improve their functional status, depression, and anxiety levels and might positively influence body image. There were no changes in caregivers’ levels of anxiety and depression.
[28] (2017) Greece RCT (brief report)—quasi-experimental? At home	Evaluate the safety and effectiveness of a home care programme for children with CF and to assess the value of regular telephone contact with the CF team. Total of 60 children and adolescents. Mean age 13.25 ys (±2.62). Teleconsultation group: 34 patients living 60–400 km from the hospital; home visit group: 26 patients living within 60 km of the hospital.	Technological requirements: telephone. Intervention design: telephone communication with a home care team, which included questions on exacerbation, respiratory infections or symptoms, weight gain, medications, and treatment adherence. Frequency: twice a week and whenever patients needed it.	No significant statistical differences were found between the two groups in FEV1 and the days and cost of hospitalization after the implementation of the home care program. The study showed that home care was a highly effective option that improved QoL, treatment adherence, and lung function among patients with CF and also recommended the use of regular telephone communication to increase treatment adherence, especially among patients with CF living far from the CF centre.
[29] (2021) Brazil Cross-sectional study At home	Describe the experience of implementing routine teleconsultations on respiratory physiotherapy at a reference centre for CF during the COVID-19 pandemic. Total of 184 participants—137 children and 47 adolescents. Mean age 7.2 ys (±5.3). Teleconsultation group: n = 153, 71 males and 82 females, mean age 7.0 ys (±0.5); in-person group: n = 31, 16 males and 15 females, mean age 8.0 ys (±1.0).	Technological requirements: web-based platform (Skype^®^) or telephone, depending on the patient’s availability. Intervention design: The teleconsultations were multidisciplinary, with the various specialties covering the treatment of CF (pulmonology, physical therapy, nutrition service, nursing, and social service). The appointment was made by the pulmonology service and the sequence of care provided was organised together with the professionals, so that everyone could carry out individual and sequential teleconsultations with the patients. Physical therapy telemedicine was divided into two segments: teleconsultation and telemonitoring. The topics covered in the physical therapy teleconsultation included treatment adherence, the proper use of prescribed drugs, the possibilities of performing techniques, and the use of physical therapy equipment for each patient individually. Thus, the possibility of open dialogue between the physical therapist and the family was offered, addressing important issues of the daily treatment routine, with the possibility of adapting it during the COVID-19 pandemic. Frequency: telemedicine was available every three months to all patients followed up at the referral centre.	Most children and adolescents with CF participated in teleconsultations and adhered to them, which demonstrated the importance of remote care activities during the period of the COVID-19 pandemic. There were no statistical differences between the group followed by teleconsultation and the group followed in person, regarding the rates of bacterial colonization of the respiratory tract and FEV1. This care strategy was considered positive by the multidisciplinary team.

## Data Availability

The original contributions presented in the study are included in the article; further inquiries can be directed to the corresponding author.

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
