# Peer review of "Telerehabilitation in Children and Adolescents with Cystic Fibrosis: A Scoping Review"

_healthcare, 2024, doi:10.3390/healthcare12100971_

Round 1

Reviewer 1 Report

Comments and Suggestions for Authors Thank you for inviting me to review this manuscript. Telerehabilitation is an emerging and important  part of rehabilitation currently and this review addresses an important aspect of this field. I have the below comments: What were the participants’ age groups included in the review? It is mentioned in the methods, but  also this should be reported in the introduction to set the scene for the reader. Study design: Authors mentioned that they searched in several databases, which databases  authors checked/searched? The flow of the results is nice, and is well written, but I believe that authors should, at least, partly, report the findings of the included studies regarding the outcome measures to strengthen their  review. What were the results of searching grey literature? This should be reported. Comments on the Quality of English Language

Minor Edits 

Reviewer 2 Report

Comments and Suggestions for Authors

Dear Respectable Authors

Thank you for considering a significant area of research related to telerehabilitation, especially in patients with cystic fibrosis. Your manuscript is well-designed and your results are of interest. However, the way you report the manuscript needs some revisions as follows. I hope my recommendation will better the quality of your manuscript.

- Following the journal guideline, please remove subheadings from the abstract section.

- Lines 21-2, I think it is better to add more details here. These statements are not clear. Please add databases, search period, and exact data of search. Also, there is a difference between a systematic review and a scoping review, and in my opinion, scoping reviews are not a type of systematic review. Please refine these statements. Also, add more details regarding eligibility criteria, study selection, data charting, and synthesis.

- Lines 22-3, the first result of all reviews is the number of the included studies, please mention it in the abstract. Also, if your results are frequent, add them here in front of each intervention. For example, how many final included studies refer to video games? If you have such information, enter it as a percentage or frequency.

- Abstract, conclusion you should provide a direct and clear answer to the question of your study in the conclusion, regardless of statistical difficulties for the general readers.

- Line 87, systematic or scoping?

- Please follow the PRISMA ScR as you mentioned. Please take a look at items on this checklist, item 6 is the first item. You must first specify the eligibility criteria and then select keywords and search databases.

- In my opinion, lines 102-9 are redundant. 

- Please add a separate heading for "information sources and search" and also for "study selection" (items 7, 8, and 9 of PRISMA ScR).

- What is your search period? it is unclear.

- Only two duplicate results in the first step. It is not rational. Please check it again. You searched 4 main databases and 2 duplicates are not rational. 

- Line 173, this statement is redundant and is related to the method section. 

- The reasons for exclusion at the final stage are not clear.

- Your final results in not based on the subheadings that you provided in the abstract. What is the reason? Please restructure your results in three sections; 1) Selection of sources of evidence, 2) Characteristics of sources of evidence, and 3)Results of individual sources of evidence.

- Line 315, this statement is completely wrong. Look at item 12 of PRISMA ScR. It is not mandatory for such types of studies because it is assumed that in this type of review, the number of final included articles is very high, on the other hand, since no specific analysis is done, there is no need for mandatory evaluation. However, due to the small number of studies included in your review, I recommend that you remove this limitation and add the quality assessment of the studies to increase the consistency of your work.

Cheers

Comments on the Quality of English Language

There are some types of punctuation and grammatical errors in the text that need edit by a native language editor.

Reviewer 3 Report

Comments and Suggestions for Authors

Introduction:

Different body systems, including the respiratory system, are affected by CF. Authors should start broad and then specify respiratory system as most affected by CF.

Authors should define children and adolescents age groups.

I recommend that authors paraphrase the CF introduction (combine the CF definition in your protocol with the definition in the manuscript).

1.1. Objectives

Since, according to your statement, to date no systematic reviews were conducted on the use of telehealth in the provision of rehabilitation care – telerehabilitation – in children and adolescents with CF, why authors have chosen to conduct a Scoping review not a systematic review?

Methods:

Have authors searched Cochrane library to check for systematic reviews related to the review question?

Study design:

Line 145, line 156 and line 173: Add the initials of the two authors names who screened the titles … etc

Figure 1- PRISMA: the figure is not clear for the reader. Add boxes and lines/arrows if appropriate.

Line 192-193: ‘’All studies were conducted at patients’ homes.’’. Paraphrase to ’’ All studies were conducted as home-based interventions’’.

Authors should update their search, currently it is 2017-2022

Results:

The presentation of the results can be enhanced.

The results are presented in a way with no outcome of results are reported. I understand the nature of the review question, but reporting the outcome of the results is important. These should be linked to the discussion.

Authors should report the outcomes of the studies, including the adherence rates-if available-.

Comments on the Quality of English Language

Minor edits

Round 2

Reviewer 2 Report

Comments and Suggestions for Authors

Dear Respected Authors

Thank you for your clarification. In my opinion, the revised manuscript is acceptablein this fashion.

Cheers

Reviewer 3 Report

Comments and Suggestions for Authors

Authors addressed my comments.

Comments on the Quality of English Language

-